# Visual Acuity and Contrast Sensitivity in Preterm and Full-Term Children Using a Novel Digital Test

**DOI:** 10.3390/children10010087

**Published:** 2022-12-31

**Authors:** María Teresa Pérez Roche, Jason C. Yam, Hu Liu, Diego Gutierrez, Chau Pham, Victoria Balasanyan, Gerardo García, Mauricio Cedillo Ley, Sandra de Fernando, Marta Ortín, Victoria Pueyo

**Affiliations:** 1Ofthalmology Department, Miguel Servet University Hospital, 50009 Zaragoza, Spain; 2Aragon Institute of Heatlh Research (IIS Aragón), 50009 Zaragoza, Spain; 3Hong Kong Eye Hospital, Hong Kong, China; 4The First Affiliated Hospital of Nanjing Medical University, Nanjing 210029, China; 5I3A Institute for Research in Engineering, Universidad de Zaragoza, 50009 Zaragoza, Spain; 6National Institute of Ophthalmology, Hanoi 100000, Vietnam; 7Yasny Vzor Idt., 107031 Moscow, Russia; 8Strabismus and Pediatric Ophthalmology Department, Hospital de la Ceguera, APEC, Ciudad de Mexico 04030, Mexico; 9Ophthalmology Department, Cruces University Hospital, 48903 Barakaldo, Spain; 10DIVE Medical S.L., 50018 Zaragoza, Spain

**Keywords:** visual acuity, contrast sensitivity, preferential looking, eye tracking, preterm birth

## Abstract

Visual assessment in preverbal children mostly relies on the preferential looking paradigm. It requires an experienced observer to interpret the child’s responses to a stimulus. DIVE (Device for an Integral Visual Examination) is a digital tool with an integrated eye tracker (ET) that lifts this requirement and automatizes this process. The aim of our study was to assess the development of two visual functions, visual acuity (VA) and contrast sensitivity (CS), with DIVE, in a large sample of children from 6 months to 14 years (y) of age, and to compare the results of preterm and full-term children. Participants were recruited in clinical settings from five countries. There were 2208 children tested, 609 of them were born preterm. Both VA and CS improved throughout childhood, with the maximum increase during the first 5 years of age. Gestational age, refractive error and age had an impact on VA results, while CS values were only influenced by age. With this study we report normative reference outcomes for VA and CS throughout childhood and validate the DIVE tests as a useful tool to measure basic visual functions in children.

## 1. Introduction

A main milestone in childhood is the acquisition of verbal language, which allows expressing inner senses as vision. Before this natural boundary, we can only access knowledge about a child’s vision by observing their behaviour. There are many tests based on the observation of the natural child’s responses to visual stimuli. However, this places excessive importance on the figure of the observer and his or her experience, and therefore visual tests may lack objectivity.

For instance, the preferential looking paradigm is the tendency to fix the gaze on a patterned stimulus over a plain one and is present from early childhood. Traditionally, paediatric ophthalmologists test vision by presenting two stimuli simultaneously and observing how the child fixates the gaze on one of them, if they are capable, to see it [1]. Other methods to measure visual acuity (VA) or contrast sensitivity (CS) usually require asking the subject to recognize and name aloud particular stimuli. Therefore, these procedures are restricted to verbal and cooperative patients, excluding the youngest children and people with certain disabilities. This is unfortunate, since the first approach to measuring a child’s vision is always through the measurement of both VA and CS; together, they profile the global visual performance of a child.

Digital devices, on the other hand, have improved many fields in healthcare, offering increased accuracy and reliability in the results. However, their adoption is not uniform across different clinical areas, due to price, size, or ease of use, among others. Visual examination in children is one of these fields.

DIVE (Device for an Integral Visual Examination, DIVE Medical SL, Spain) is a digital device designed to measure and analyse different key aspects of the visual function in children. DIVE is based on eye-tracking technology, which records the position of the gaze on the screen. It presents different stimuli to the patient and records the reaction of the children’s eyes, thus allowing to objectively evaluate visual function in an automated and accurate way, independent of interpretations by the examiner. Moreover, it adapts the test in real time to the child’s responses and behaviour, allowing for more precise measurements.

Premature children are at risk of abnormal visual development since their infancy. Therefore, finding an early and accurate method for their visual assessment is even more critical among these children. Their visual problems can affect several areas, including ocular motility, visual fields or visual perception; however, the first approach to a child’s vision is always through the measurement of VA. Along with the CS, both can profile a child’s global visual performance.

The aim of the present study was to describe the development of these two abilities throughout childhood and to compare them between a cohort of preterm and born-at-term children using a DIVE device.

## 2. Materials and Methods

### 2.1. Study Design

The present study is part of the TrackAI project that used DIVE to assess visual functions for screening purposes [2]. DIVE obtained values of VA, CS, oculomotor control and colour perception from a large sample of children aged 6 months to 14 years (y). The study was conducted between April 2019 and May 2021 in tertiary hospitals and ophthalmologic clinics in five countries: Spain, China, Vietnam, Russia and Mexico. It is therefore a cross-sectional, observational study. The study was registered with the number ISRCTN17316993.

### 2.2. Population

There were two study groups: children born at term and with normal vision (Norm) and preterm children (Pre). The children were recruited consecutively from patients attending the centres during the recruitment period. To be included in the study, all children and/or their parents or tutors had to sign the informed consent form.

Inclusion criteria for the Norm group:Age between 6 months and 14 y.No relevant medical reports (including gestational age at birth ≥ 37 weeks (w) and birthweight ≥ 2500 gr).Low refractive error: myopia equal of less than 3.50 dioptres (D) for children younger than 30 months, 3.00 D between 31 and 48 months and 1.50 D over 48 months; hyperopia equal or less than 4.50 D for children younger than 30 months, 4.00 D between 31 and 48 months and 3.50 D over 48 months; astigmatism equal to or less than 2.00 D for children younger than 48 months and 1.50 D over 48 months [3].Labelled by a paediatric ophthalmologist as “normal visual development” after a complete visual examination.

Exclusion criteria for the Norm group:Refractive error higher than specified above, ocular surgery, strabismus or any other ophthalmologic disease.General diseases directly or potentially affecting visual performance (i.e., central nervous system disorders, intellectual disabilities, psychological problems, endocrine disorders).General poor health to perform the ophthalmologic examination and the digital test.

Inclusion criteria for the Pre group:Age between 6 months and 14 y.Gestational age < 37 w.VA enough to fixate on a small picture (3 mm) at 40 cm distance.

Exclusion criteria for the Pre group:General poor health to perform the ophthalmologic examination and the digital test.

### 2.3. Clinical Protocol

All included patients underwent a complete ophthalmologic examination and DIVE tests. Monocular assessment was attempted in all the children. This was performed by patching with a soft eye patch the non-tested eye. When a child did not cooperate, the exam was performed binocularly. For comparative reasons, the results report the right eye in case of monocular testing.

#### 2.3.1. Ophthalmologic Examination

The ophthalmologic examination, conducted by a paediatric ophthalmologist, included the following procedures:Subjective evaluation of oculomotor control: fixation, smooth pursuit movements and saccadic performance.Distance VA without optical correction, using the appropriate method according to the child’s age:

<2 y: Lea gratings, version #253300 (Good-Lite Vision, Elgin, IL, USA). This was performed at a distance of 57 cm, under binocular vision and following the test instructions.

>2 to 4 y or illiterate: Lea symbols, version #250220 (Precision Vision Ltd., La Salle, IL, USA) on a printed card measured at 3 m distance.

>5 y: ETDRS, charts 1 and R (Precision Vision Ltd., La Salle, IL, USA) on a printed card measured at 3 m distance. Prior to the test, the examiner checked the recognition of the Latin script of the children from Russia, China and Vietnam.

For both Lea symbols and ETDRS, there had to be three or more symbols or letters correctly identified in a row, to obtain a certain VA level.
Monocular near VA without optical correction measured at 65 cm, with printed LEA card for near VA in cooperative children.Examination of ocular alignment and motility, using Hirschberg or cover tests.External ocular examination with slit lamp or lantern.Refraction under cycloplegia—measured by automated refraction or retinoscopy.Funduscopic examination.

These last two exams were performed after the DIVE test to avoid the effect of cycloplegia and mydriasis on visual function.

An ophthalmologist conducted all the above examinations. After that, the doctor labelled the child’s visual function as “normal visual development” or “abnormal visual development”.

#### 2.3.2. DIVE Testing

DIVE is a digital device with a high-resolution screen and an integrated eye tracker (ET). The system was made up of a Huawei Matebook E tablet with a 12-inch tactile screen corresponding to a visual angle of 27.07 deg horizontally and 14.82 deg vertically from a distance of 65 cm, with a resolution of 2160 × 1440 pixels, as well as an integrated X3-120 Tobii ET sampling at 120 Hz. The display was regularly calibrated with a Datacolor SpyderX calibrator (gamma 2.20, white dot 6500 K, and 120 cd/m^2^). The manufacturer’s specifications for the binocular accuracy and precision of the ET are around 0.6° and 0.25°, respectively (0.8° and 0.34° for the monocular case).

The study was conducted in a quiet room with mesopic illumination. The child was seated on their own or on the parent’s lap in front of DIVE, at 65 cm distance. Two ellipses on the screen serve as a guide for the correct eye position for the test. First, the device starts a calibration process in which the child has to fixate and follow a cartoon image on the screen through nine different positions uniformly distributed across the screen (Figure 1). Each individual point was repeated, if necessary, until the ET reported a reliable calibration. In addition, a subsequent validation test allowed us to quantify the ET’s exact accuracy and precision for each subject.

After that, the test shows four circles with different luminance, against a plain grey background. One of the circles has a pattern of stripes of a certain width or contrast (Figure 2). Following the principle of preferential looking, if the child is able to discriminate the stripes, he or she will fixate on the striped circle instead of the homogeneous greys. DIVE will offer positive feedback if the patient looks at the striped stimulus within three seconds, by showing a star in the centre of the circle and a reward sound. A correct answer will make the following trial more difficult to recognize, which means, in the VA test, narrower stripes in the patterned circle and, in the CS test, less contrast difference between the stripes. In case the child does not discriminate the pattern and looks at another circle, the next trial will be easier to recognise.

DIVE uses a psychophysical adaptive method to present the stimuli. The range of stimuli in grating VA starts at 0.25 cycles per degree (cpd) and reaches up to 18 cpd. The CS test was performed with stimuli of 0.5 cpd. The range of contrasts displayed starts at 99% of contrast (0.0 uLog) up to 0.3% (3 uLog).

After the tests, apart from the visual outcomes, DIVE scored the quality and reliability of each exam based on several internal parameters. The test had to achieve at least 3 points on a scale of 1 to 5 to be included.

### 2.4. Statistical Analyses

Statistical analyses were carried out with the statistical software SPSS 21.0 (SPSS Inc., Chicago, IL, USA). Student’s *t*-test, chi-square test or ANOVA were used to compare the outcomes between the study groups and age ranges. A *p* value lower than 0.05 was considered statistically significant.

Normative curves were created using the GAMLSS (Generalized Additive Models for Location, Scale and Shape) package in R [4]. The Akaike information criteria [5] and Q-test were used to evaluate goodness of fit. The models for all the variables included parameters that account for skewness and kurtosis in the distribution of the values. Additionally, age-adjusted VA and CS tolerance limits were reported, as the range in which 90 percent of a normally distributed population is found, with 95% probability.

Regression models were performed including age at the study, gestational age at birth, sex, ethnicity, previous retinopathy of prematurity and refractive error as independent variables and VA or CS as dependent variables. The strength of the effect of each independent variable on visual outcomes was compared by the standardized beta coefficient (b_std_).

## 3. Results

### 3.1. Testability

Out of 2208 tested children, 97% (2134 children) were able to complete the VA test, and 89.13% (1968 children) met the quality criteria to be included. There were significant differences in excluded children between study groups, with a higher percentage of excluded children in the Pre group than in the Norm group (17.8% vs. 8.45%, *p*< 0.001). Regarding CS, 96% (2118 children) completed the test, while 87.41% (1930 children) obtained reliable data and were included. Exclusion rates were also higher in the Pre group (19.87% vs. 9.82%, *p* < 0.001). VA test was performed binocularly by 307 children (15.5%) with a mean age of 2.15 y (standard deviation (s. d) 1.55). CS test was performed binocularly by 229 children (11.8%) with a mean age of 2.27 y (s. d 1.66).

Mean times to complete the DIVE tests are 50 s for the initial calibration, 31.4 s for the VA test and 35.9 s for the CS test. 

### 3.2. Norm Group

Norm group consisted of 1599 healthy children without any medical issues, including perinatal outcomes. The average age at the time of testing was 6.20 y and the sex distribution was 50% female. Included children were geographically and ethnically diverse: African black, Indian, Latin American, Middle East, Oriental and White. The white group of children was the largest, with 47.6% of the sample. Refractive measurements were within the range of normal vision at 65 cm (Table 1).

All included participants were divided into five groups based on developmental stages of childhood: infancy (<1 year, group 1), toddlerhood (1–2 years, group 2), early childhood (3–5 years, group 3), middle childhood (6–11 years, group 4) and adolescence (≥12 years, group 5).

The results of VA and CS improved throughout childhood, with significant differences among age groups for both functions (Table 2). However, the maximum increase was observed during the first 5 y, as shown in Table 2 and Figure 3 and Figure 4.

### 3.3. Pre Group

There were 609 preterm children in the Pre group. Age was corrected for gestational age, for better comparison of the youngest children. The mean age at inclusion in the study was 5.01 y, which means that these children were younger than those included in the Norm group at the time of testing. The percentage of females was 45% and the proportion of white children was 72.6% (Table 3). There were significant differences in sex distribution and ethnicity between the two study groups. In addition, all characteristics based on the inclusion criteria (i.e., gestational age, birth weight) and those related to them (such as refraction or prevalence of visual disorders) also differed between the groups. Preterm children increased VA and CS throughout childhood, with significant differences among age ranges (Table 4).

### 3.4. Norm Curves

Reference curves for VA and CS outcomes as a function of age were built, obtained from the values of the Norm group, with defined thresholds in 10th and 90th centiles. For enabling comparisons between study groups, a reference line representing the median of the values from the Pre group has been plotted over the normative curves. VA norm curve is shown in Figure 3.

The same procedure was followed to build CS norm curves (Figure 4).

### 3.5. Multivariate Analysis

A regression model was built for both dependent variables, VA and CS, considering sex, ethnicity, gestational age, retinopathy of prematurity, age and refractive error as independent variables. This last variable was calculated as the absolute value of the spherical equivalent of the right eye (for children examined monocularly) or the mean spherical equivalent of both eyes (for children examined binocularly). As expected, ethnicity did not influence VA or CS values. For VA, age (βstd = 0.299, *p* <0.001), gestational age (βstd = 0.106, *p* = 0.003), refractive error (βstd = −0.134, *p*< 0.001) and sex (βstd = 0.068, *p* = 0.033) were included in the model and reached statistical significance. In the regression model for CS, only age (βstd = 0.476, *p* < 0.001) was statistically significant.

## 4. Discussion

The visual system is characterized by marked immaturity at birth, followed by progressive improvement in morphological and functional aspects. Retinal specialization and myelinisation of visual pathways are the main anatomical changes. This physiological maturation enables improved VA and other functions.

Establishing the boundaries of VA in young children remains a challenge, as values are highly dependent on the method applied [6], conditioned in turn by attention, cognitive capacity and even the interaction of the examiner.

Traditionally, methods for visual testing in young children were divided into objective and subjective measurements. The first group is mainly based on visual evoked potentials while subjective methods are all behavioural techniques that are partially dependent on the observer’s criteria. DIVE was developed with the aim of automatizing measurements using ET technology. DIVE detects eye responses, overcoming the need for a qualified observer and minimizing the patient’s collaboration requirements. The digital stimuli and the automatic and adaptative presentation of the stimuli avoid the source of bias of examiners’ lack of experience and objectivity and increase the repeatability of the tests. In addition, prior to the present project, DIVE reported good correlation values with the Lea grating test in children younger than four, in a recently published validation study [7].

A further classification can be made within VA tests according to the stimuli presented. Grated stimuli reflect the resolution acuity (the ability to discriminate high-contrasted stripes), and letters, symbols and other figures are known as recognition methods, in which the patient has to identify the symbol and be able to communicate it. Some studies have shown that grating acuity and letters or symbols correlate well in collaborative patients [8], but for younger children other studies showed a tendency of grating tests to overestimate VA [9,10] or to underestimate it [11]. Therefore, direct comparison between visual tests is not always possible [12]. Anstice warned about comparing different VA methods in children, especially when the task changes, i.e., from resolution to recognition [13]. The present study has provided insights into the development of two visual functions using the same method in a large group of children from 0.5 y to 14 y, which minimizes the effect of using different tests.

Despite the great diversity of methods, there is some agreement on the general periods of VA development [14]. There is a rapid increase in VA between six months and the first year of life, with a steady but slower improvement thereafter, so that adult levels are reached around the ages of five to eight years [6,15,16,17]. Our results partially agree with those developmental stages. In our sample, there was an improvement in the mean VA throughout the first five years, but the values continued to increase until 12 y and older children. Values of grating acuity in children aged 0–4 y had already been established with subjective techniques in large population samples [18,19]. These studies focused on early visual development, with age groups for each month from birth. Our results from the first group (<1 y) obtained a mean VA similar to the reported values for 9–12 months by other authors. The trend of VA in older children is similar to other studies, with the greatest increase during the 6–7 years of life [14], which is relevant for clinical practice. However, when the study was conducted, DIVE had a range of VA gratings from 0.25 cpd to 18 cpd and caused a ceiling effect at 18 cpd that mainly affected older ages.

Defining Normal CS boundaries provides valuable information since there are many visual conditions that may alter this function [20]. Its measurement has two components: the size of the displayed pattern and the differences in luminance. Based on both properties, we can depict the complete range of CS in the population and represent it as a function: the contrast sensitivity function (CSF). The way this function changes throughout childhood has been the focus of many studies. Although there is not complete agreement on how and when the CSF changes, there is evidence for a greater increase in sensitivity at low frequencies throughout childhood [21,22,23]. The present study only evaluated one low frequency, 0.5 cpd. It meets the goals of better detection of age-related changes, ensuring that all ages, even younger children, can see the frequency shown, and for screening purposes, requiring shorter test times [24]. In addition, CS at high spatial frequencies correlated well with VA, given that it requires high contrast to be seen.

Our results showed a large improvement of CS over ages. Children older than 13 y had a sensitivity three times better than those younger than 1 y, with the greatest increase during the first 5 y of life. Similarly, another study found that CS levels in adults were double the values in toddlers, with the greatest increase before 4 y [22]. It should be noted that the range of displayed contrasts in DIVE is wider than that of old analogic methods [25], which experienced roof effects. Digital CS tests can display, and therefore obtain, higher CS values than printed tests [26], so that former normal values of CS should be revised. As CS has been described as “the most complete single measure of human spatial vision” [22], there is a need for tools which combine accurate testing and clinical utility.

The normal ranges of VA and CS were wide in our sample, with the largest differences in the group of children between 3 and 5 y. This may reflect the influence of attention on any assessment in toddlers. At these ages, the preferential looking paradigm is not so instinctive as in infants, and the ability to maintain attention or follow instructions is not yet well developed. Therefore, as other authors have observed using ET, sometimes negative results (lower VA than expected) are obtained that may be mediated by the child’s attention [10]. A way to decrease this effect is to repeat the test until the child pays attention, but our protocol, as based in a clinical setting, did not allow it. However, it is important to notice that the shape of the development in VA and CS in our sample has nearly the same profile, with their maximum increase during the first 5 y. Our previous local results with DIVE in normal children confirm this tendency [7].

Preterm children face a higher rate of visual problems throughout their lives. Prospective cohort studies from different countries address more visual problems in all age stages, from newborn to school-age children and young adults. In adulthood, people who were born prematurely seem to have lower VA, CS or visual field sensitivity [27]. Apart from the known risk of retinopathy with prematurity, premature children face a higher rate of multiple visual disorders throughout childhood, such as strabismus, refractive errors or cerebral visual disorders [28]. However, the presence of refractive errors or other visual disorders alone do not fully explain the lower VA scores, as shown by regression models adjusted for these variables.

Several studies found better VA values in preterm compared to full-term children [29] and related this to their early visual experience. However, longer follow-ups led to the conclusion that they have a lower VA [30], although only a small percentage have mild or severe visual impairment [28]. Our results found a lower mean and median VA in preterm children, especially those in older than 7 y. We may suspect a difficulty in reaching higher levels of VA that persists into late childhood. In this regard, Hellgren reported lower VA in teenagers [31], and Petursidorff, in young adults born preterm [27].

Fewer studies have evaluated CS in premature children, as CS tests for children are scarce. There are two main types of studies: infants tested with the subjective preferential looking paradigm (which is highly dependent on observer’s experience, as mentioned above) and older children assessed at cooperative ages with adult tests. An example of the latter is the study by Larsson in preterm and full-term children at the age of 10. It found differences in all spatial frequencies tested, wider in children with a previous retinopathy of prematurity. Conversely, DIVE CS results showed no effect of gestational age on this function. This may be due to differences in the frequencies tested (0.5 cpd in DIVE, from 1.5 to 18 cpd in Larsson’s study) or even in the group of premature children, since they had lower GA and higher rate of retinopathy of prematurity. However, other authors found no differences in CS between preterm and full-term infants when testing was performed during the first year of life [32,33,34]. Long-term studies found similar results in young adults, a preserved CS, despite an affected VA [35]. In conclusion, CS in preterm children still lacks developmental studies using accurate and comparable methods throughout childhood.

Premature children have higher rates of significant refractive errors [27,28,29,30,31,35], in many cases related to the treatment of the retinopathy of prematurity [36]. However, therapeutic advances, such as intravitreal anti-VEGF therapy, have reported promising results in decreasing the rate of refractive errors [37]. In our sample, the Pre group had higher refractive errors than the Norm group, and VA rather than CS was actually affected more prominently.

The use of eye-tracking technology to measure VA or CS leads to the automatization of the preferential looking paradigm. Several studies confirmed the usefulness of this method, both in CS in adults [38] and in VA in preverbal children [9]. DIVE tests were completed in 96–97% of the children, similar to previously reported results using ET in children (97–100%) [9,10]. Only results with good quality and reliability were included in the study, amounting to 87.4–89.1%. Due to our clinical protocol, the quality of the tests was only checked during the analysis of the data. For this reason, tests with poor quality were not repeated. Reviewing this parameter in real time and repeating suboptimal tests would have for sure increased the rate of high-quality exams. However, these are good testability ratios, considering that our results come from a nearly real clinical practice, with different settings, ages and observers, and a very large number of included patients (70 times greater than the studies referenced above). Therefore, our study takes the next step in the application of ET technology for visual exploration by using it for large population samples. In addition, we agree with Holmqvist on the need for reporting data quality and post-recording exclusion criteria in ET studies [39].

Recently, other eye-tracker-based devices have demonstrated their utility in detecting strabismus and other risk factors for amblyopia [40,41]. We plan to perform more studies in the future to explore the global capacity of DIVE as a vision screener.

We can mention some limitations and strengths of the present work. First, the number of recruited children differ among the age groups. Since all the children fulfilling the inclusion criteria attending one of the participating centres were included in the study, differences are due to clinical and demographic characteristics of the centres. As this is a cross-sectional and observational study, we analysed children who are developing well enough to attend ophthalmology consultations from 6-months-old onwards, so that our sample of premature children may lack those with the worst progression. Furthermore, our records of birth weight were not calculated according to gestational age, so we missed the information related to the small-for-gestational-age children.

Due to the difficult access to clinical data in certain countries, some of the medical information was collected through parent’s reports, which may lead to inaccuracy in some of the diagnoses, especially at perinatal period. Therefore, aspects as important as treated ROP were not accurately collected and consequently not reported in the present study. Preterm and term-born groups differed among them in certain demographic outcomes, such as sex rates or ethnicity. Our higher rate of males in the preterm group responds to the global distribution of prematurity, as reported by many other authors [42]. As far as ethnicity, differences may be due to clinical diversity between the recruitment centres.

The TrackAI project focuses on vision screening, therefore the tests did not include the entire CS function that implies assessing contrast sensitivity at different spatial frequencies on a grating pattern, but only the CS value at 0.5 cpd. The highest grating acuity examined in the DIVE test was 18 cpd, which had a ceiling effect in older patients. This technical limitation has already been solved and current versions of the test allow the assessment of grating VA up to 29 cpd.

One of the main strengths of the present study is the psychophysical method behind the presentation of the stimuli in DIVE, which is adaptative in real time to the child’s responses and minimizes the duration of the test for a predefined precision.

To our knowledge, this study reports the assessment of VA and CS in one of the largest and most diverse samples of premature and term-born children, using the same objective method. Other studies based on the preferential looking paradigm have the critical limitation of having different examiners judging the child’s visual behaviour, with the added difficulty of making comparisons between subjective tests. In addition, the number of included countries provides a global view and overcomes particular biases related to the children’s origin.

In conclusion, with this study we provide insights into the development of two main visual functions, visual acuity and contrast sensitivity, throughout childhood and evaluate the influence of prematurity, and other related events, on these skills.

## Figures and Tables

**Figure 1 children-10-00087-f001:**
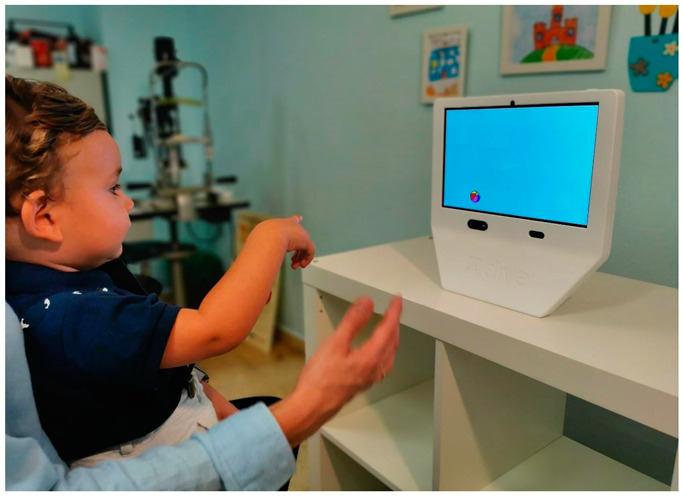
A child performing the calibration process of DIVE.

**Figure 2 children-10-00087-f002:**
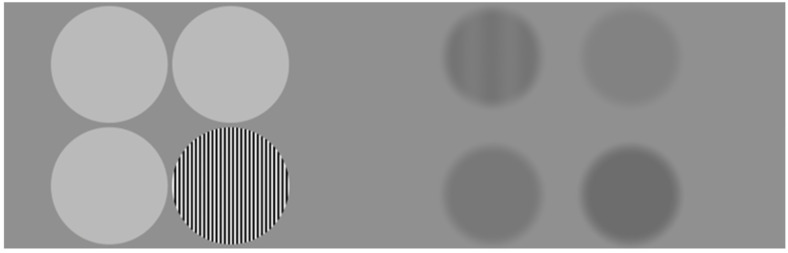
Example screen during the VA test (**left**) and the CS test (**right**).

**Figure 3 children-10-00087-f003:**
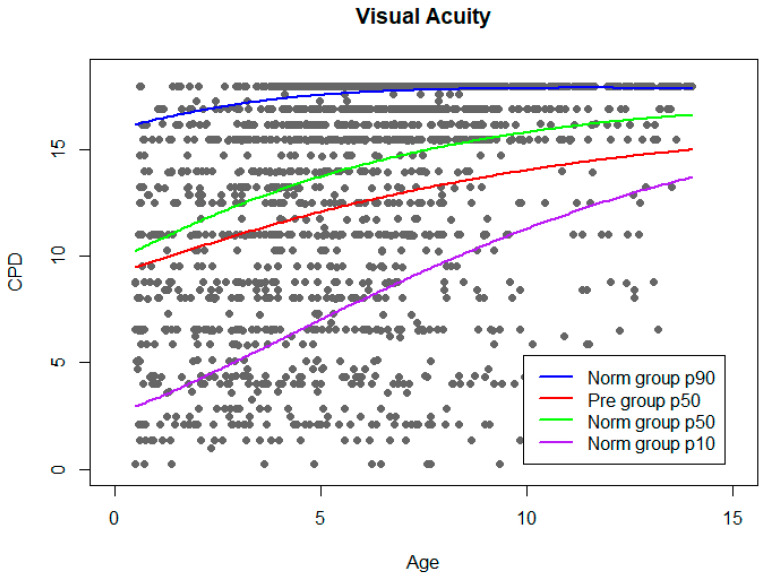
VA norm curve. Grey dots represent VA results from Norm group. Adjusted normal percentiles (10, 50, and 90) are shown as colour lines. Percentile 50 of Pre group was plotted on the graph.

**Figure 4 children-10-00087-f004:**
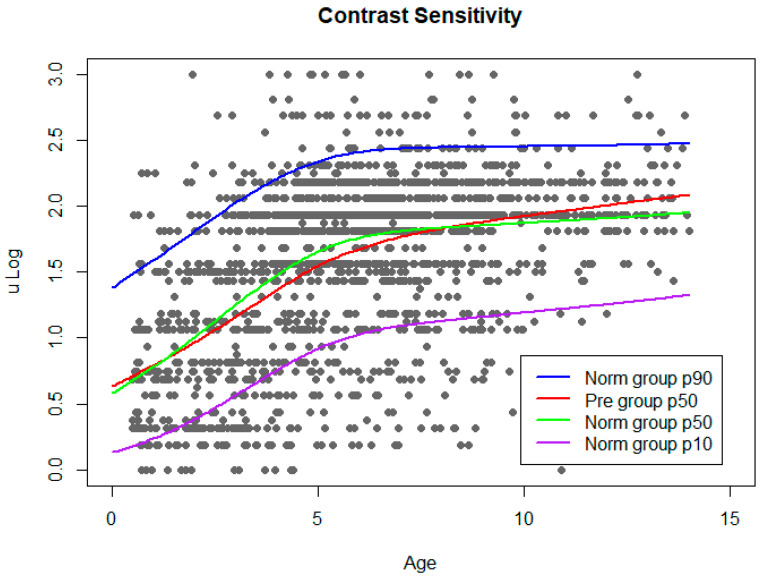
CS norm curve. Grey dots represent CS results from Norm group. Adjusted normal percentiles (10, 50, and 90) are shown as colour lines. Percentile 50 of Pre group was plotted on the graph.

**Table 1 children-10-00087-t001:** Norm group results.

	Mean	s. d	Range
Age (y)	6.20	2.87	0.5–13.99
Gestational age (w)	39.46	1.00	38–44
Birth weight (gr)	3330.58	412.15	2501–5000
RE Cycloplegic refraction sph (D)	+1.02	1.08	−3.00–+4.25
RE Cycloplegic refraction cyl (D)	−0.13	0.72	−2.00–+2.00
LE Cycloplegic refraction sph (D)	+1.06	1.07	−2.00–+4.50
LE Cycloplegic refraction cyl (D)	−0.13	0.74	−2.00–+2.00

RE: right eye, LE: left eye, sph: sphere, D: diopters, cyl: cylinder, s. d: standard deviation.

**Table 2 children-10-00087-t002:** AV and CS results in the Norm group, according to age groups.

		<1 y	1–2 y	3–5 y	6–11 y	>12 y	*p*
VA (cpd)		N = 33	N = 135	N = 526	N = 706	N = 63	
	Mean (s.d)	8.80 (4.59)	9.87 (5.10)	13.94 (4.80)	15.11 (4.33)	16.54 (3.24)	<0.001 *
CS (u log)		N = 29	N = 133	N = 520	N = 699	N = 61	
	Mean (s.d)	0.74 (0.59)	0.99 (0.62)	1.66 (0.53)	1.82 (0.47)	1.89 (0.50)	<0.001 *

* ANOVA between age groups. Data are presented as mean (s. d).

**Table 3 children-10-00087-t003:** Pre group results.

	Mean	s. d	Range
Age (y)	5.01	3.27	0.50–13.79
Gestational age (w)	31.52	3.50	23–36
Birth weight (gr)	1741.87	749.73	480–4400
RE Cycloplegic refraction sph (D)	+1.38	2.29	−12.00–+9.00
RE Cycloplegic refraction cyl (D)	+0.03	1.50	−6.00–+5.00
LE Cycloplegic refraction sph (D)	+1.40	2.29	−10.50–+9.00
LE Cycloplegic refraction cyl (D)	+0.02	1.55	−6.00–+4.50

RE: right eye, LE: left eye, Sph: sphere, D: diopters, cyl: cylinder, s.d: standard deviation.

**Table 4 children-10-00087-t004:** AV and CS results in the Pre group, according to age groups.

		<1 y	1–2 y	3–5 y	6–11 y	≥12	*p*
VA (cpd)		N = 42	N = 101	N = 188	N = 151	N = 23	
	Mean (s.d)	8.07 (4.85)	9.74 (4.86)	12.00 (5.15)	13.10 (5.19)	13.54 (5.27)	<0.001 *
CS (u log)		N = 34	N = 97	N = 186	N = 149	N = 22	
	Mean (s.d)	0.71 (0.51)	0.99 (0.59)	1.51 (0.71)	1.79 (0.54)	2.07 (0.46)	<0.001 *

* ANOVA between age groups. Data are presented as mean (SD).

## Data Availability

The data associated with the paper are not publicly available, but they are available from the corresponding author upon reasonable request.

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
