# Peer review of "Visual Acuity and Contrast Sensitivity in Preterm and Full-Term Children Using a Novel Digital Test"

_children, 2022, doi:10.3390/children10010087_

Round 1

Reviewer 1 Report

Summary of Paper: IRB consented, 5-country (Spain, China, Vietnam, Russia and Mexico) 4/19 to 5/21 (COVID period) multicentered evaluation of new Spanish eye-tracking video vision test on 2208 children, 1599 “norm” mean age 6.2±2.9 years BW 3330 grams  and 307 ex-premature children mean age 5.0 ± 3.3 years BW 1740 grams.  47 % ethnic “white.”  Refractive error not extreme, but small astigmatism due to excludion of cyl greater than 2 diopters..  In the middle of an eye exam with attempted age-appropriate grating, LEA or ETDRS acuity, The digital eye tracking grating acuity first calibrated and if 3 of 5 scored, then acuity and contrast sensitivity (CS)  with 0.5 CPD estimated.  The device uses grating acuity in a four-way presntation rather than 2-Way Teller but eye tracking to score, with a ceiling of 18 CPD limiting the teen age precision.  Cycloplegia was used for refraction and retinal exam AFTER DIVE.  97% completed VA test and 89% completed constrast sensitivity.  Monocular testing was not done in 307 children mean age 2.2 years VA and 229 children mean age 2.3 years CS. 

For normal sbjects the estimated range of grating acuity ranged form 12 to 18 CPD for older children, and from 4 to 16 in younger children with the median slightly worse for the premature born children.  Contrast ranged from 1.2 to 2.5 µLOG for older children and from 0.3 to 1.5 µLOG for younger children similar for premature born.

The age-related distribution of estimates for VA were impaced multivariable analysis most by age, gestational age, refractive error and gender, but for CS only age impacted.

No correlation between exam visual acuity and DIVE was performed to validate the device.

Ophthalmologist sorted into “normal” or abnormal development, but these groups not analyzed differently.

 The psychophysical method tried to minimize the duration of the test.

If additional reports are planned to cover other aspects, Please report this.

Major Comments:  

The data was obtained, but no comparision of exam visual acuity and DIVE visual acuity estimate made.  This is very important.

Did non-English children have matching card for SLOAN letters not it their alphabet.  (ETDRS) or were different ETDRS optotypes used?

What consituted endpoint on each test (ie 3 of 4 correct indicated passing line?).

The estimated DIVE visual acuity should be compared to exam visual acuty using Intraclass correlation and Bland Altman analysis.

Line 175: Please further define “psychophysical adaptive” method. 

Was time-of-testing measured?  From the software?  It would be helpful data point.

Does this study have a registry ( please lis in Methods) and de-identified data available? 

Eyetracking has been used is some other forms of pediatric vision screening, not just grating acuity (your reference 8 for instance).  Please add to your discussion:

(Mahlen T, Arnold R. Pediatric non-refractive vision screening with EyeSwift, PDI Check and blinq. Clin Ophthalmol. 2021;16:375-384. doi:10.2147/OPTH.S344751

And 

Yehezkel O, Spierer A, Oz D, Yam R, Belkin M, Wygnanski-Jaffe T. An objective rapid sytem based on eye tracking for eye deviation measurement in children. Invest Ophthalmol Vis Sci. 2018;59:1024.)

Minor Comments:  

Line 163- “grey” more UKL english while “gray” mor American english- your choice.

Table 4 and Table 2- describe what “N” means since you do not have an “=” sign between N and 42.  

Line 244 built

Line 116- mention that the “monocular assessment” is for the DIVE, not the exam.  Then tell us how monocularity was achieved- did you patch the non-tested eye?

Line 124: Why were visions tested without correction?

Line 134- Instead of “collaborative” you may prefer “cooperative”

Line 391 surely instead of “for sure”  and these is a good, should be there is a good, or these are a good.

Line 81.  Instead of transversal (intersecting lines), you might intend to say “international” or “multi-Ethnic” or “Multi-centric”?

Author Response

Reviewer 1

Comments and Suggestions for Authors

Summary of Paper: IRB consented, 5-country (Spain, China, Vietnam, Russia and Mexico) 4/19 to 5/21 (COVID period) multicentered evaluation of new Spanish eye-tracking video vision test on 2208 children, 1599 “norm” mean age 6.2±2.9 years BW 3330 grams  and 307 ex-premature children mean age 5.0 ± 3.3 years BW 1740 grams.  47 % ethnic “white.”  Refractive error not extreme, but small astigmatism due to excludion of cyl greater than 2 diopters..  In the middle of an eye exam with attempted age-appropriate grating, LEA or ETDRS acuity, The digital eye tracking grating acuity first calibrated and if 3 of 5 scored, then acuity and contrast sensitivity (CS)  with 0.5 CPD estimated.  The device uses grating acuity in a four-way presntation rather than 2-Way Teller but eye tracking to score, with a ceiling of 18 CPD limiting the teen age precision.  Cycloplegia was used for refraction and retinal exam AFTER DIVE.  97% completed VA test and 89% completed constrast sensitivity.  Monocular testing was not done in 307 children mean age 2.2 years VA and 229 children mean age 2.3 years CS.

For normal sbjects the estimated range of grating acuity ranged form 12 to 18 CPD for older children, and from 4 to 16 in younger children with the median slightly worse for the premature born children.  Contrast ranged from 1.2 to 2.5 µLOG for older children and from 0.3 to 1.5 µLOG for younger children similar for premature born.

The age-related distribution of estimates for VA were impaced multivariable analysis most by age, gestational age, refractive error and gender, but for CS only age impacted.

No correlation between exam visual acuity and DIVE was performed to validate the device.

Ophthalmologist sorted into “normal” or abnormal development, but these groups not analyzed differently.

 The psychophysical method tried to minimize the duration of the test.

If additional reports are planned to cover other aspects, Please report this.

A: Thank you for time reviewing our work, you perfectly summarized the article. In relation to this last point, as the aim of TrackAI project was to screen visual function, we will report global screening performance in the subsequent studies.

Major Comments: 

The data was obtained, but no comparision of exam visual acuity and DIVE visual acuity estimate made.  This is very important.

A: In a recently published paper, we compared DIVE VA test and Lea grating test, as a validation study. There were good correlation values between both tests.  Esteban-Ibañez, E., Perez-Roche, T., Prieto, E., Castillo, O., Fanlo-Zarazaga, A, et al. Age norms for grating acuity and contrast sensitivity in children using eye tracking technology. Int Ophthalmol [Internet]. 2021 [cited 2021 Nov 3]; Available from: https://pubmed.ncbi.nlm.nih.gov/34622374/

Did non-English children have matching card for SLOAN letters not it their alphabet.  (ETDRS) or were different ETDRS optotypes used?

A: All the literate children that completed ETDRS were able to answer without matching letters, including children from Vietnam, Russia and China. This skill was confirmed before performing clinical tests.

What consituted endpoint on each test (ie 3 of 4 correct indicated passing line?).

A: Yes, this was the criterion in ETDRS and LEA Symbols. We added the information in the main text.

The estimated DIVE visual acuity should be compared to exam visual acuty using Intraclass correlation and Bland Altman analysis.

A: Addressed above

Line 175: Please further define “psychophysical adaptive” method.

A: Details of the psychophysical method are protected under trade secret.

Was time-of-testing measured?  From the software?  It would be helpful data point.

A: We added mean testing times at the end of the “Testability” subsection, within Results, with the following sentence: “Mean times to complete the DIVE tests are 50 s for the initial calibration, 31.4 s for the VA test, and 35.9 s for the CS test.”

Does this study have a registry ( please lis in Methods) and de-identified data available?

A: We registered the study with the number: ISRCTN17316993

Further details: https://www.isrctn.com/ISRCTN17316993?q=ISRCTN17316993&filters=&sort=&offset=1&totalResults=1&page=1&pageSize=10

We currently haven’t a public repository. We can give access to the database necessary to reproduce our results under request.

Eyetracking has been used is some other forms of pediatric vision screening, not just grating acuity (your reference 8 for instance).  Please add to your discussion:

(Mahlen T, Arnold R. Pediatric non-refractive vision screening with EyeSwift, PDI Check and blinq. Clin Ophthalmol. 2021;16:375-384. doi:10.2147/OPTH.S344751

And

Yehezkel O, Spierer A, Oz D, Yam R, Belkin M, Wygnanski-Jaffe T. An objective rapid sytem based on eye tracking for eye deviation measurement in children. Invest Ophthalmol Vis Sci. 2018;59:1024.)

A: We added these valuable references as an example of future lines of study.

Minor Comments: 

Line 163- “grey” more UKL english while “gray” mor American english- your choice.

A: We changed all words to “grey”.

Table 4 and Table 2- describe what “N” means since you do not have an “=” sign between N and 42. 

A: Requested change has been applied to the tables

Line 244 built

A: Changed, thank you.

Line 116- mention that the “monocular assessment” is for the DIVE, not the exam.  Then tell us how monocularity was achieved- did you patch the non-tested eye?

A: Yes, we added this information to the main text.

Line 124: Why were visions tested without correction?

A: Because the main purpose of the study was visual screening.

Line 134- Instead of “collaborative” you may prefer “cooperative”

A: Changed, thank you.

Line 391 surely instead of “for sure”  and these is a good, should be there is a good, or these are a good.

A: Changed, thank you.

Line 81.  Instead of transversal (intersecting lines), you might intend to say “international” or “multi-Ethnic” or “Multi-centric”?

A: We meant “cross-sectional”. We changed it in the text.

"Please see the attachment".

Reviewer 2 Report

I believe it is a scientifically sound research that should validate the DIVE and I am looking forward to be able to use DIVE in clinical practice. I do have several minor questions/concerns:

line114:" When a child did not cooperate [...] the exam was performed binocularly." Since presumably binocular VA is better than monocular VA, it should be at least mentioned what was the percentage of children that did not cooperate for monocular VA testing.

line 163: "DIVE will offer positive feed back" How long should the child fixate (presumably seconds) before the positive feed back is given?

lines 287-291 Even if you make your argument against comparisons between different methods of VA testing, a correlation coefficient between DIVE and Lea gratings (for example) would provide useful information for the reader

Tables 2 and 4 Please specify that the "N" letter stands for the number of children tested in each group

Author Response

Reviewer 2

Comments and Suggestions for Authors

I believe it is a scientifically sound research that should validate the DIVE and I am looking forward to be able to use DIVE in clinical practice. I do have several minor questions/concerns:

A: Thank you for the time revising our work. In the following we address the comments raised in the reviews.

line114:" When a child did not cooperate [...] the exam was performed binocularly." Since presumably binocular VA is better than monocular VA, it should be at least mentioned what was the percentage of children that did not cooperate for monocular VA testing.

A: This aspect is addressed at the end of the “Testability” subsection, within Results. “VA test was performed binocularly by 307 children (15.5%) with a mean age of 2.15 y (standard deviation (s. d) 1.55). CS test was performed binocularly by 229 children (11.8%) with a mean age of 2.27 y (s. d 1.66).”

line 163: "DIVE will offer positive feed back" How long should the child fixate (presumably seconds) before the positive feed back is given?

A: We added this information as follows: “DIVE will offer positive feedback if the patient looks at the striped stimulus within three seconds.” More details on these algorithms are protected under trade secret and not disclosed to users.

lines 287-291 Even if you make your argument against comparisons between different methods of VA testing, a correlation coefficient between DIVE and Lea gratings (for example) would provide useful information for the reader

A: Prior to this project, we reported and published the comparison between DIVE VA test and Lea grating test, with good correlation results. You can find the paper here: 26. Esteban-Ibañez, E., Perez-Roche, T., Prieto, E., Castillo, O., Fanlo-Zarazaga, A, et al. Age norms for grating acuity and contrast sensitivity in children using eye tracking technology. Int Ophthalmol [Internet]. 2021 [cited 2021 Nov 3]; Available from: https://pubmed.ncbi.nlm.nih.gov/34622374/

Tables 2 and 4 Please specify that the "N" letter stands for the number of children tested in each group

A: We added an equal (=) sign to clarify the meaning

"Please see the attachment".
